

# Validation of a general subjective well-being factor using Classical Test Theory

Ali Al Nima[1,2], Kevin M. Cloninger[1,3], Franco Lucchese[4], Sverker Sikström[5] and Danilo Garcia[1,2,6]

[1] Blekinge Center of Competence, Region Blekinge, Karlskrona, Sweden
[2] Department of Psychology, University of Gothenburg, Gothenburg, Sweden
[3] Anthropedia Foundation, St. Louis, Missouri, USA
[4] Department of Dynamic and Clinical Psychology, University of Rome "La Sapienza", Rome, Italy
[5] Department of Psychology, Lund University, Lund, Sweden
[6] Department of Behavioral Sciences and Learning, Linköping University, Linköping, Sweden

Corresponding authors
Ali Al Nima,
alinor_1979@yahoo.co.uk
Danilo Garcia,
danilo.garcia@icloud.com

## ABSTRACT

**Background:** Subjective Well-Being (SWB) is usually conceptualized in terms of an affective (i.e., judgements of biological emotional reactions and experiences) and a cognitive component (i.e., judgements of life satisfaction in relation to a psychological self-imposed ideal). Recently, researchers have suggested that judgements of harmony in life can replace or at least complement the cognitive component of SWB. Here, however, we go beyond that suggestion and propose that harmony in life should be seen as SWB's social component since it is the sense of balance between the individual and the world around her—a process that comprises acceptance, adaptation, and balance. By adding judgements of one's social interactions (i.e., harmony in life) to judgments of one's life satisfaction (psycho) and judgements of one's emotional reactions (bio), we propose a tentatively biopsychosocial model of SWB. As a first step, we used different factorial models in order to determine if both a general factor and specific sub-factors contribute to the biopsychosocial model of SWB.

**Method:** A total of 527 participants responded to the Positive Affect Negative Affect Schedule (PANAS; 20 items), the Satisfaction with Life Scale (SWLS; five items), and the Harmony in life Scale (HILS; five items). We conducted exploratory and confirmatory factor analyses to validate the biopsychosocial model of subjective well-being and a general factor (SWBS).

**Results:** The 20 PANAS items reflected a mixture of general latent structure saturation and specific latent structure saturation, but contributed to their respective specific latent factor (PA: 48%; NA: 49%) more than to the general latent SWBS factor (positive affect: 25%; negative affect: 32%). The five SWLS items contributed to a larger degree to the general SWBS factor (72%) than to life satisfaction itself (22%), while the five HILS items contributed to even a larger degree to the general SWBS factor (98%) than to harmony in life (0%). The bifactor model was the best model compared with all other models we tested ($\chi^2 = 1{,}660.78$, df = 375, $p < 0.001$); Satorra Bentler $\chi^2 = 1{,}265.80$, df = 375, $p < 0.001$; CFI = 0.92; Tucker–Lewis Index = 0.91; RMSEA = 0.067. This model of a general SWBS factor explained about 64% of the total variance in the model, while specific SWBS components together explained 15% of the total variance.

**Conclusion:** Our study suggests SWB as a general factor in a multidimensional biopsychosocial model. Indeed, as much as 64% of the variance of SWB was explained by this general factor. The SWB components, however, contributed to a different degree to each corresponding factor in the model. For instance, while the affective and cognitive components seem to be their own constructs and also part of the general SWB factor, the social component tested here contributed 0% to its own variance but 98% to the general factor.

## INTRODUCTION

*The whole is greater than the sum of its parts.*

Aristotle

*I . . . a universe of atoms, an atom in the universe.*

Richard P. Feyman

For over 35 years, subjective well-being has been conceptualized as a dyad consisting of an affective and a cognitive component/part (*Diener, 1984*). The affective component is often conceptualized as one's evaluations of biological positive and negative emotional experiences in daily life, while the cognitive component is often conceptualized as evaluations of one's life as a whole in relation to a psychological self-imposed ideal—together, the frequent experience of positive affect, the infrequent experience of negative affect, and being satisfied with life is the way researchers operationalize high subjective well-being (*Diener et al., 1985*, *2009*; *Cloninger, 2004*; *Lyubomirsky, 2008*; *Peterson & Seligman, 2004*). In this configuration, subjective well-being is considered as one of the best available proxies for a broader, more canonical form of well-being (*Diener, Lucas & Oishi, 2018*) and strongly associated with personality[1] (*Eid & Larsen, 2008*). Indeed, well-being depends on a person's ability to achieve *homeostasis* or to adapt to circumstances outside the self and to characteristics within the person (*Cloninger, 2004*). In this context, some researchers suggest that subjective well-being needs to also be understood as inner harmony (*Delle Fave et al., 2011*, *2016*), while others suggest that harmony in life is a complement to or even a supplement of the cognitive component of subjective well-being—life satisfaction (*Kjell et al., 2016*). We agree in the fact that this component is extremely important for our understanding of subjective well-being, since harmony in life is the expression of a process of acceptance and adaptation in order to achieve inner peace and balance or *homeostasis* within and outside the self. We argue, however, that harmony in life is distinct to life satisfaction, especially in light of a biopsychosocial perspective on subjective well-being.

The biopsychosocial model is a scientific model that refers to a dynamic and complex interaction of physiological, psychological, and social factors that can both result in and

[1] Personality can be defined as the "dynamic organization within the individual of the psychobiological systems by which the person both shapes and adapts uniquely to an ever-changing internal and external environment" (*Cloninger, 2012*).

[2] The Greek word *psyche* found in psychology and psychiatry stands for "life, soul, or spirit,", which is distinct from *soma*, which refers to the "body" (*Cloninger, 2004*; see also *Cloninger & Cloninger, 2011a, 2011b*; *Cloninger, Salloum & Mezzich, 2012*).

contribute to health (*Cloninger, 2004*; *Engel, 1980, 1977*). Such a model covers all the parts that compose a human being (i.e., body, mind, and psyche[2]), it corresponds to a ternary model of human awareness: the self, others, and something greater than the self, such as, nature, God or the universe (*Cloninger, 2004*), and corresponds also to the concept of health as a state of physical, mental, and social well-being (*WHO, 2001*). Naturally, we propose that the evaluation of positive and negative affect is the biological part of subjective well-being, since emotions are derived from our nervous system and our temperament, a part of personality with a strong genetic factor that is relatively stable over the life span (*Cloninger, 2004*; *Josefsson et al., 2013*; *Zwir et al., 2018a, 2018b, 2019*). In contrast, although also relatively stable, the cognitive component of subjective well-being, life satisfaction, seems to fluctuate with time and to be influenced by changes in life circumstances (*Fujita & Diener, 2005*), such as, divorce and losing one's life partner (*Lucas et al., 2003*). Since life satisfaction is an evaluation of one's life in relation to a psychological self-imposed ideal, we propose that it should be understood as the psychological part of a biopsychosocial model of subjective well-being. As a result of this suggestion, the question is the how harmony in life is different from life satisfaction?

Firstly, as a concept, harmony is related to the sense of balance and flexibility that a person experiences in relation to her life and the world around her (cf. *Li, 2008*). Ergo, conceptually, it involves transcendence of the self and comprises notions of a person being in balance, in agreement, or striving for equilibrium with the environment (e.g., surroundings, other people, family, friends, nature, and her own existence). Secondly, psychometrically, despite a strong correlation between life satisfaction and harmony in life ($r = 0.76$), two-factor model solutions, rather than single factor models, seem considerably better when researchers use the Satisfaction with Life Scale and the Harmony in life Scale to operationalize these constructs (*Kjell et al., 2016*). That being said, harmony in life is distinctive from life satisfaction, not only due to how it is conceptualized or psychometric differences between measures, but also because the meaning of the words people use to describe how they pursue harmony is semantically different from the words people use to describe how they pursue life satisfaction (*Kjell et al., 2016*). For instance, people use more frequently words such as *peace, balance, unity, agreement, calm, mediation, cooperation, tolerant, nature, forgiveness*, etc., when describing how they pursue harmony in life vs. when describing how they pursue life satisfaction (*Kjell et al., 2016*). Conversely, people use more frequently words such as *job, money, achievement, education, success, wealth, house, gratification*, etc., when describing how they pursue life satisfaction vs. when describing how they pursue harmony in life (*Kjell et al., 2016*). Therefore, we go beyond suggesting harmony as a complement to the cognitive component of subjective well-being and propose that, from a biopsychosocial perspective, harmony in life should be seen as the social component of a general subjective well-being factor.

By adding judgements of one's social interactions (harmony in life) to judgements of one's emotional reactions (bio) and judgments of one's life satisfaction (psycho), we used Classical Test Theory (CTT) to investigate different factorial models of our theorized biopsychosocial general subjective well-being factor and its specific sub-factors. Next, we
briefly review the current literature that have addressed each component using CTT. Here we only review the most common measure for each of these components (for a compilation of measures see, for example, *Lopez & Snyder, 2003*).

## Three components, three measures

The Positive Affect Negative Affect Schedule (*Watson, Clark & Tellegen, 1988*) has been used in several studies to assess the affective or biological component of subjective well-being. This instrument consist of 20 items, 10 adjectives that measure positive affect (i.e., "Interested", "Enthusiastic", "Proud", "Alert", "Inspired", "Determined", "Attentive", "Active", "Excited", and "Strong") and 10 adjectives that measure negative affect ("Distressed", "Upset", "Guilty", "Afraid", "Hostile", "Irritable", "Ashamed", "Nervous", "Jittery", and "Scared") with a 5-point Likert scale (1 = *not at all*, 5 = *very much*). The best representation of positive and negative affect's latent structure is the orthogonal rotation of the factors, perhaps due to the opposing pleasant–unpleasant relationship in the factor loadings (*Watson, Clark & Tellegen, 1988*). The scales have shown high internal consistency in different studies—Cronbach's alphas raging between 0.83 to 0.90 for positive affect and between 0.85 to 0.93 for negative affect (see *Watson & Clark, 1994*; *Leue & Lange, 2011*). Nevertheless, researchers have reported a two-factor model with positive affect and negative affect as both uncorrelated factors and correlated factors (*Crawford & Henry, 2004*; *Kercher, 1992*; *Mackinnon et al., 1999*; *Terraciano, McCrae & Costa, 2003*; *Killgore, 2000*; *Mehrabian, 1997*; *Ortuño-Sierra et al., 2015*, *2019b*; *Sanmartín et al., 2018*). Moreover, using structural equation modeling, the best-fitting models are achieved by specifying correlations between error in items closely related to each other in meaning: Distressed-Upset, Guilty-Ashamed, Scared-Afraid, Nervous-Jittery, Hostile-Irritable, Interested-Alert-Attentive, Excited-Enthusiastic-Inspired, Proud-Determined, and Strong-Active (*Crawford & Henry, 2004*). Hence, these covariances partially suggest the possibility of item reduction without serious repercussions on the internal consistency reliability of the positive and negative affect scales (*Thompson, 2007*, *2017*).

The Satisfaction with Life Scale is a brief assessment of the cognitive or psychological component of subjective well-being (*Diener et al., 1985*; *Pavot & Diener, 1993*; *Pavot & Diener, 2008*; *Glaesmer et al., 2011*; *Moksnes et al., 2014*; *Ortuño-Sierra et al., 2019a*). The scale consists of five items (i.e., "In most ways my life is close to my ideal", "The conditions of my life are excellent", "I am satisfied with my life", "So far I have gotten the important things I want in life", and "If I could live my life over, I would change almost nothing") with a 7-point Likert response scale (1 = "*strongly disagree*" to 7 = "*strongly agree*"). The Satisfaction with Life Scale has shown Cronbach's alphas ranging from 0.79 to 0.89 (e.g., *Pavot & Diener, 1993*; *Adler & Fagley, 2005*; *Steger et al., 2006*; for a meta-analysis see *Vassar (2008)*). Moreover, in the original article (*Diener et al., 1985*), a principal-axis factor analysis on the Satisfaction with Life Scale resulted in a single factor solution, in which the single factor accounted for 66% of the variance in the items. Although the single factor solution has been replicated in several studies, the fifth item (i.e., "If I could live my life over, I would change almost nothing") often shows lower factor loadings and lower item-total correlations than the first four items (e.g., *Senécal, Nouwen & White, 2000*).

Probably because this specific item clearly implies an evaluation over one's whole past life, while the other items of the scale imply a focus on the present (e.g., "The conditions of my life are excellent") or a temporal summation (e.g., "In most ways my life is close to my ideal") (*Pavot & Diener, 2008*).

The Harmony in Life Scale (*Kjell et al., 2016*) comprises 5 items (i.e., "My lifestyle allows me to be in harmony", "Most aspects of my life are in balance", "I am in harmony", "I accept the various conditions of my life", and "I fit well with my surroundings") with a 7-point Likert response scale (1 = "*strongly disagree*" to 7 = "*strongly agree*") and similar instructions as the Satisfaction with Life Scale. The factor loadings range from 0.73 to 0.90 (e.g., *Kjell et al., 2016*; *Singh, Mitra & Khanna, 2016*) and Cronbach's alphas from 0.83 to 0.95 in different studies (e.g., *Kjell et al., 2016*, *2019*; *Garcia, Nima & Kjell, 2014*; *Singh, Mitra & Khanna, 2016*). CTT studies show that despite a strong correlation between life satisfaction and harmony in life, the two-factor models, rather than single factor models, are considerable better (*Kjell et al., 2016*).

## The present study

Our aim was to investigate different factor models of our theorized biopsychosocial model of subjective well-being, and its general factor and specific sub-factors: positive and negative affect (bio), life satisfaction (psycho), and harmony in life (social). We suggest that seeing subjective well-being from a biopsychosocial perspective covers all the parts that compose a human being (i.e., body, mind, and psyche), it also corresponds to a ternary model of human awareness: the self, others, and something greater than the self, such as, nature, God or the universe (*Cloninger, 2004*), and to the concept of health as physical, mental, and social well-being (*WHO, 1946*). To the best of our knowledge, this is the first study to examine a general subjective well-being factor using *higher order factor analysis* and *Bifactor analysis*.

## METHOD

### Ethics statement

Ethics approval was not required at the time the research was conducted as per national regulations. The consent of the participants was obtained by virtue of survey completion after they were provided with all relevant information about the research (e.g., anonymity).

### Participants and data collection procedure

The participants ($N = 600$ in the initial sample, with an age mean of 39.41 sd = 12.43; in which 74.60% were employed for wages and 79.50% had a Bachelor's degree as their highest achieved educational level) were recruited through Amazon's Mechanical Turk[3] (http://www.mturk.com/mturk/welcome). All participants originated from the USA and spoke English as their first language. Participants were informed that the survey was voluntary, anonymous, that they could terminate the survey at any time and that those who accepted would receive $0.50 as compensation for their participation. We added two control questions to the survey, to control for automatic responses (e.g., This is a control question, please answer "neither agree nor disagree"). The final sample, after taking

---

[3] Amazon's Mechanical Turk allows data collectors to recruit participants (i.e., workers) online for completing different tasks for money (for a review on the validity of this method for data collection see among others: *Buhrmester, Kwang & Gosling, 2011*).

away those who responded erroneously to one or both of the control questions ($N$ = 73; 12.17% of all respondents) consisted of 527 participants (200 males and 327 females).

## Measures

**Positive Affect and Negative Affect Schedule (PANAS;** *Watson, Clark & Tellegen, 1988*) The PANAS instructs participants to rate to what extent they generally have experienced 10 positive, PA (e.g., "Proud"), and 10 negative, NA (e.g., "Afraid"), feelings and moods during the last week, using a 5-point Likert scale (1 = *very slightly or not at all*, 5 = *extremely*).

The Satisfaction with Life Scale (SWLS) (*Diener et al., 1985*) assesses the cognitive component of subjective well-being (i.e., life satisfaction) and consists of five items (e.g., "In most of my ways my life is close to my ideal") that require a response on a 7-point Likert scale (1 = *strongly disagree*, 7 = *strongly agree*).

Harmony in Life Scale (HILS) (*Kjell et al., 2016*) assess a person's global sense of harmony in life and consists of 5 statements (e.g., "My lifestyle allows me to be in harmony") for which respondents are asked to indicate degree of agreement on a seven-point Likert scale (1 = *strongly disagree*, 7 = *strongly agree*).

## Statistical treatment

The Expectation-Maximization Algorithm (EM-Algorithm) was used to deal with missing values (less than 0.8% participants in all variables/items). Little's Chi-Square test for Missing Completely at Random was, $\chi^2$ = 590.64 (df = 637, $p$ = 0.91). This means that the missing data was missing at random and not systematically; thus, the EM-Algorithm was appropriate for replacing the missing data (*Tabachnick & Fidell, 2007*). Based on the EM-Algorithm, 527 participants' responses (males = 200, females = 327) were found to be valid. All items had skewness between −0.01 to 2.35 and kurtosis between −0.15 to −2.77 except the items "guilty", "hostile" and "ashamed" with kurtosis = 4.01, 4.26 and 5.26 respectively. The values of skewness and kurtosis regarding these items indicated that we have violated the assumption of normality, so we used Satorra Bentler $\chi^2$ in our analysis for model goodness of fit vs. the null (independence) model (*Tabachnick & Fidell, 2007*). This test is a correction to the chi-squared test and makes standard errors, $p$-values, and confidence intervals robust to nonnormality. The items were averaged to compute SWLS, HILS, NA and PA and then added to compute a subjective well-being total score (SWBS). When we computed the scores and Cronbach's alphas of SWBS we reversed the scores of the NA items. In other words, the raw score of these items are subtracted rather than added in the computations of SWBS because the items are negatively related to the SWBS construct. Items of the four subscales (SWLS, HILS, NA and PA) had different ranges, so we also standardized them before we computed the SWBS average. This was done to make sure that all items contribute equally to SWBS and to make it easier to interpret the results. The scales (SWBS, SWLS, HILS, NA and PA) had skewness between −0.19 to 1.65 and kurtosis between 0.06 to 2.06. See the Supplemental Material for the details (Table S1).

## Statistical procedure

We used the following software to analyze the data: STATA version 14, SPSS version 24 and Microsoft Excel. As a first analysis, we tested and described the map of correlations among all items (30 items) in our study, and the correlations among SWBS (simple average of standardized scores of all items including reserved scores of NA items) and its four specific subscales (SWLS, HILS, NA and PA) using Pearson's correlation coefficient (Table S2). In the second analysis, we investigated convergent and discriminant validity by conducting a Person correlation analysis between SWLS, PA, NA, HILS.

As a third analysis, our main statistical procedures were: (1) to run exploratory factor analyses (EFA) using principal component analysis (PCA) to describe and cover the underlying structure regarding proposed models of the scales (SWBS, SWLS, HILS, NA and PA) as five separate latent traits, (2) to run confirmatory factor analyses (CFA) using structural equation modeling (SEM) with *robust* maximum likelihood (ML) (*Satorra & Bentler, 2001*) estimation to test our theoretical model and to determine whether measures of each construct are consistent with earlier understandings of the nature of underlying factorial structure (latent trait) of these scales (SWBS, SWLS, HILS, NA and PA). We applied five different EFA and CFA models: (a) to investigate if the correlation among items in each specific subscale were explained by only a single latent trait (four separate unidimensional factor structures), (b) to test that the correlations among all 30 items were dependent on only a single general latent trait (unidimensional factor structure of SWBS), (c) to test the proposed multidimensional correlated model of the subscales (SWLS, HILS, NA and PA) without a single general latent trait (SWBS), (d) to test a higher order multidimensional factor model (second order model) using SWBS as single general latent factor and a second order factor and using the subscales (SWLS, HILS, NA and PA) as specific first order factors and domains/traits, (e) to test a bifactor model (that can be considered as a nested factors or hierarchical factor model) to investigate the proposed multidimensional factor structure of SWBS and its subscales, in which every item is affected by both SWBS as a single general latent trait and by and only by its respective subscale (SWLS, HILS, NA and PA) as orthogonal secondary dimensions.

As a fourth set of analyses, we tested the scales reliability using both Cronbach's alpha reliability coefficient and Omega reliability coefficients. We used Cronbach's alpha coefficient to evaluate the internal reliability of each subscale and the overall scale of SWBS. All scales in our study had high reliability, with Cronbach's alpha ranging from 0.92 for PA to 0.96 for SWBS. See the Supplemental Material for the details (Table S3). This high internal reliability could, however, be explained due to the fact that Cronbach's alpha coefficient is influenced by different sources (i.e., general, group, and specific factors). Moreover, the high internal reliability, as measured by Cronbach's alpha, might be just a reflection of the reliability of all these sources without partitioning. Importantly, Cronbach's alpha is based on observed variances and covariances and assumes that all items have equal loadings on the latent factors, hence, it depends on the average item intercorrelation and the number of items in the scale (i.e., as the number of items and the intercorrelation values increase, so does Cronbach's alpha) (*Rodriguez, Reise & Haviland, 2016*). As the matter of

fact, Cronbach's alpha may indicate high internal reliability even when the data reflects highly multiple latent structures, or it can also underestimate internal reliability when the data has a unidimensional latent structure. Indeed, when the data has a multidimensional structure, Cronbach's alpha is affected by all sources of common and specific item variance, and it can over- or under-estimate the reliability of the scale(s). Moreover, in this case, it is not entirely straightforward what the correct interpretation of "true score" variation is because the "true score" itself also is a weighted composite of multiple latent dimensions (*Reise, Bonifay & Haviland, 2018*). With these limitations of Cronbach's alpha in mind, we also computed Omega reliability coefficients, which have high generalizability and could also help us to avoid the limitations of the Cronbach's alpha coefficient. *Omega* coefficients are based on factor loadings, do not assume equal loadings, and can separate out the reliable item's variance to either latent general factors or latent subscales factors. In other words, Omega calculations take into consideration both general and group sources of common variance as "true score" variance and estimates the reliability of a multidimensional scale (*Reise, Bonifay & Haviland, 2018*). In our study we computed five Omega coefficients. (1) *Omega total* ($\Omega Total$), which is a reliability estimate that provides information of the amount of the common variance of items' variance in a given model that belongs to a reliable variance for the general latent factor and also for specific subscale(s). (2) *Omega hierarchical* ($\Omega H$), which is a reliability estimate that provides information about the amount variance of all items in a given model that belongs only to a reliable variance for the general latent factor. (3) *Omega hierarchical subscale* ($\Omega HS$), which is a reliability estimate that provides information of the amount of the subscale reliable variance after controlling for reliable variance due to the general latent factor. (4) *Omega subscale* ($\Omega S$), which is a reliability estimate that provides information of the amount of the subscale reliable variance due to both the general latent factor and the specific corresponding subscale. Finally, (5) *Omega general for subscale*, which is a reliability estimate that provides information of the subscale reliable variance in a given model accounted only by the reliable variance of the general latent factor.

As a fifth analysis, we computed explained common variance (ECV). The ECV is a statistical reliability index that provides information of the percentage of the common variance explained by the general latent factor. In the sixth analysis, we computed item explained common variance (I-ECV). The I-ECV is a statistical reliability index that provides information of the percentage of the common variance at the item level that is expected by the general latent factor. In the last analysis, we calculated and compared the participants' scores in SWLS, PA, NA, HILS, and SWBS between the models.

## RESULTS

### The map of correlations among all items

The map of correlations among items in our study showed that most of the correlations were high and significant, ranging from −0.12, $p < 0.01$ to 0.91, $p < 0.01$. The largest correlation, both within and between subscales, was between the item "Most aspects of my life are in balance" and the item "I am in harmony" ($r = 0.91$, $p < 0.01$) and the lowest was between the item "Inspired" and the item "Guilty" ($r = −0.04$, *ns*). Some items, such as

the item "Alert", had however only low correlations to items within and outside its specific subscale. Thus, we expected that this specific item might not be a good marker of the positive affect subscale or the general latent factor of SWB. Indeed, item covariances between items of the different subscales are pure reflections of general factor saturation, whereas covariances between items within subscales reflect a mixture of general factor saturation and group factor saturation (*Zinbarg, Revelle & Yovel, 2007*). That being said, this map of correlations showed also that many items within each specific subscale are highly correlated with each other and also across the other subscales. In other words, reflecting that a multidimensional construct, in which each item is influenced by multiple latent factors, might fit our data better rather than a unidimensional latent construct, in which each item is influenced by a single latent factor.

## Convergent and discriminant validity

In order to test convergent and discriminant validity we investigated the Pearson correlations between the different scales, which ranged between from $-0.34$, $p < 0.01$ to $0.83$, $p < 0.01$. The SWLS ($r = 0.52$, $p < 0.01$) and HILS ($r = 0.55$, $p < 0.01$) were positively and significantly correlated with the PA. Conversely, the SWLS ($r = -0.51$, $p < 0.01$) and HILS ($r = -0.60$, $p < 0.01$) were negatively and significantly correlated with NA. Moreover, PA and NA were negatively and significantly correlated with each other ($r = -0.34$, $p < 0.001$). The largest correlation was between SWLS and HILS ($r = 0.83$, $p < 0.01$). Hence, there is sufficient convergent and discriminant validity between the model's four different constructs. See the Supplemental Material for the details (Table S3).

## Exploratory factor analyses and confirmatory factor analyses

As a first step we used a series of EFA and CFA that tested whether the correlation among items in each specific subscale were explained by only its single latent trait. We applied four separate unidimensional factor structure models for SWLS, HILS, NA and PA. First, we conducted four separate EFA using PCA to examine the latent structure for each subscale. An eigenvalue greater than one (eigenvalues >1) was chosen as suitable criterion to determine that a reasonably large proportion of the total variance corresponded to at least one factor. The results showed that each subscale had only one eigenvalue that was greater than one, except for PA which had two eigenvalues that were greater than one (4.08 for SWLS, 4.21 for HILS, 6.56 for NA and both 5.88 and 1.18 for PA). Because of the large difference between the first (5.88) and the second (1.18) eigenvalue regarding PA, we interpreted that the largest eigenvalue accounted for one extracted factor with a reasonably large proportion of the total variance. We also applied Horn's Parallel Analysis for the selection of the correct number of components in an exploratory factor analysis (*Horn, 1965*). Using this criterion, the number of factors to retain corresponds to the highest eigenvalues generated from the researcher's dataset in comparison to the randomly generated eigenvalues. Our result showed that all eigenvalues larger than one generated from our dataset were larger than the corresponding random percentile eigenvalues, except for one of the two PA eigenvalues larger than one (i.e., 1.18). All items were highly and positively loaded. Loadings of items ranged between 0.81 to 0.94 for SWLS, between

0.84 to 0.95 for HILS, between 0.76 to 0.88 for NA and between 0.72 to 0.85 for PA. The only exception was the item "Alert" in the PA subscale, with a relativity lower loading (0.60).

Secondly, we applied four CFA using SEM to examine separately the unidimensional factor structures of each subscale. Regarding SWLS, the result showed that the chi-square value was significant ($\chi^2$ = 45.30, df = 5, $p$ < 0.001), Satorra Bentler $\chi^2$ was significant (S–B $\chi^2$ = 26.38, df = 5, $p$ < 0.001), the comparative fit index (CFI) was 0.99, Tucker–Lewis index (TLI) was 0.98 and the root mean square error of approximation (RMSEA) was 0.09. However, the *chi-square* statistic is heavily influenced by sample size (*Kline, 2010*), with larger samples leading to larger value and therefore, a larger likelihood of being significant. Thus, all indices indicated that the model fit was acceptable (cf. *Bollen, 1989*; *Browne & Cudeck, 1993*). All the standardized regression loadings between SWLS and its items were significant at $p$ < 0.001 (ranging from 0.87 to 0.94) with the exception of the item "If I could live my life over, I would change almost nothing", which had a loading of 0.74. See the Supplemental Material for the details (Fig. S1). For instance, this specific item often shows lower factor loadings and item-total correlations than the first four items (e.g., *Senécal, Nouwen & White, 2000*). Probably due to the fact that this item clearly implies an evaluation over one's whole past life, while the other items of the SWLS imply a focus on the present or a temporal summation (*Pavot & Diener, 2008*).

Regarding HILS, the results showed that the chi-square value was significant ($\chi^2$ = 87.65, df = 5, $p$ < 0.001), Satorra Bentler $\chi^2$ was significant (S–B $\chi^2$ = 47.41, df = 5, $p$ < 0.001), CFI was 0.98, Tucker–Lewis index (TLI) was 0.96 and RMSEA was 0.13. Thus, while the comparative fit index and Tucker–Lewis index generally indicated good model fit but the root mean square error of approximation indicates that the model was not a good-fitting model. All the standardized regression loadings between this scale and its items were significant at $p$ < 0.001 (ranging from 0.86 to 0.96) with the exception of the item "I accept the various conditions of my life", which had loading 0.76. See the Supplemental Material for the details (Fig. S2).

Regarding PA, the results showed that the chi-square value was significant ($\chi^2$ = 508.33, df = 35, $p$ < 0.001), Satorra Bentler $\chi^2$ was significant (S–B $\chi^2$ = 350.72, df = 35, $p$ < 0.001), CFI was 0.88, TLI was 0.84 and RMSEA was 0.13. Hence, indicating that the model fit was not acceptable. All the standardized regression loadings between this scale and its items were significant at $p$ < 0.001 (ranging from 0.70 to 0.85) with the exception of the item "Attentive" and "Alert", that had loadings of 0.64 and 0.51, respectively. See the Supplemental Material for the details (Fig. S3). Regarding NA, the results showed that the chi-square value was significant ($\chi^2$ = 520.12, df = 35, $p$ < 0.001), Satorra Bentler $\chi^2$ was significant (S–B $\chi^2$ = 233.83, df = 35, $p$ < 0.001), CFI was 0.90, TLI was 0.89 and RMSEA was 0.10. Thus, these fit indexes indicated a poor model fit. All the standardized regression loadings between this scale and its items were significant at $p$ < 0.001 (ranging from 0.71 to 0.88). See the Supplemental Material for the details (Fig. S4).

In general, high values of RMSEA indicated that the models were not good enough fitting models[4]. Moreover, these high values suggest that there are large residuals in these models, that might be caused by measurement error and/or a latent multidimensional

[4] RMSEA, however, is known to be inflated with low *df* models (*Kenny, Kaniskan & Mccoach, 2015*). Therefore, we have used multiple indices to provide different information about the model fit. Used together, these indices provide a more conservative and reliable evaluation of the model fit (*Maruyama, 1997*).

structure. If these residuals are caused by a latent multidimensional structure, we expected that it would reflect both general and specific latent factors as theorized by our biopsychosocial model of SWB.

We applied EFA and CFA to test if the correlations among all 30 items are explained by only one single latent trait (unidimensional model for SWBS). We conducted EFA using PCA to test this model. The results revealed four eigenvalues that were greater than one (eigenvalues >1). Eigenvalues were 13.63, 4.14, 2.57 and 1.14. Because the large different between the first and the other eigenvalues, we interpreted that one extracted factor accounted for a reasonably large proportion of the total variance, that is, the largest eigenvalue (13.63). Horn's Parallel Analysis was also applied and showed that the first three eigenvalues (13.63, 4.14 and 2.57) generated from our dataset were larger than the corresponding random percentile eigenvalues. All items were positively loaded except for the items of NA, which were negatively loaded. Loadings of items ranged between 0.52 to 0.85 (the only exception was the item "Alert", which had weak load 0.37). We conducted also CFA using SEM to test whether the correlations among all 30 items were explained only by a unidimensional model (i.e., SWBS). We reversed the scores of the NA items because these items yielded negative factor loadings with SWBS. The results showed that the chi-square value was: $\chi^2 = 6613.79$, df = 405, $p < 0.001$, the Satorra Bentler $\chi^2$ was: S–B $\chi^2 = 4991.18$, df = 405, $p < 0.001$, and that the CFI = 0.58, TLI = 0.55 and RMSEA = 0.15. All the standardized regression weights were significant at $p < 0.001$ (ranging from 0.40 to 0.91) with the exception of the item "Alert", which had loading 0.29. See the Supplemental Material for the details (Figs. S5 and S6). In general, low values of CFI and TLI, and RMSEA indicated that the model was not a good enough-fitting model. This high value of RMSEA suggested high large residuals in these models, that could be caused by a latent multidimensional structure that contains both general and specific latent factors. Thus, we made some modifications to get a better fitting regarding this model.

We applied CFA using SEM analysis to test the proposed multidimensional correlated model in which the items were loaded to only one of the multiple dimensions (SWLS, HILS, NA and PA). The result showed that the chi-square value was significant ($\chi^2 = 1,875.24$, df = 399, $p < 0.001$), Satorra Bentler $\chi^2$ was also significant (S–B $\chi^2 = 1,413.64$, df = 399, $p < 0.001$), and a CFI = 0.91, TLI = 0.90 and RMSEA = 0.07. Hence, indicating that the model fit was acceptable. All the standardized regression weights were significant at $p < 0.001$ (ranging from 0.74 to 0.94 for SWLS, 0.76 to 0.96 for HILS, 0.51 to 0.85 for PA and 0.71 to 0.89 for NA) and all the correlations among subscales were also significant at $p < 0.001$ (ranging from −0.38 between PA and NA to 0.88 between SWLS and HILS). This model was, however, unable to give us adequate information about the overall score (i.e., SWBS) because it contained only four correlated specific subscales. In other words, this model was not able to measure and capture a single common latent factor of subjective well-being. See the Supplemental Material for the details (Fig. S7).

We applied CFA using SEM analysis to test a higher order factor model (second order model), that represents the proposed multidimensional factor structure of SWBS and its specific subscales. In this model, the subscales are dependent on only SWBS as a single

general latent trait, and items are dependent on only one of the specific subscales. The results showed that the chi-square value was significant ($\chi^2 = 1881.49$, df = 401, $p < 0.001$), the Satorra Bentler $\chi^2$ was also significant (S–B $\chi^2 = 1419.90$, df = 401, $p < 0.001$), and a CFI = 0.91, TLI = 0.90 and RMSEA = 0.07. Thus, indicating that the model fit was acceptable. All the standardized regression weights were significant at $p < 0.001$, between SWBS and its subscales (ranging from 0.60 to 0.96), and between each subscale and their items (ranging from 0.74 to 0.94 for SWLS, 0.76 to 0.96 for HILS, 0.51 to 0.85 for PA and 0.70 to 0.89 for NA). In this model, the correlations (double headed arrows) among subscales (first order latent factors) in Fig. S8 are explained by a higher order factor (SWBS) as a single general latent factor that could account for their effect. Moreover, SWBS could directly influence its specific subscales but it could only indirectly influence each item. In sum, this model did not clearly describe the effect of a general latent subjective well-being factor on each item because these specific subscales (first order traits) mediated this effect, so there was no direct relationship (pathway) between SWBS and each item. In this model, we reversed the scores of the NA items, because these items yielded negative factor loadings with SWBS. See the Supplemental Material for the details (Figs. S8 and S9).

We applied CFA using SEM analysis to investigate the proposed multidimensional factor structure of SWBS and its subscales as a bifactor model. In this model, all items were specified to load on SWBS as a single general latent trait and on one and only one corresponding specific subscale (i.e., SWLS, HILS, NA and PA) as specific factor. Moreover, in this model, all included factors (i.e., SWBS, SWLS, HILS, NA and PA) were orthogonal and uncorrelated with one another. The fit indices were acceptable ($\chi^2 = 1660.78$, df = 375, $p < 0.001$; S–B $\chi^2 = 1,265.80$, df = 375, $p < 0.001$; CFI = 0.92, TLI = 0.91 and RMSEA = 0.067), thus, indicating that this model was the best model regarding fit indexes compared with the all other models in this study, so we considered this model in our further analysis. All the standardized regression weights were significant at $p < 0.001$, except the loadings from HILS to its specific items, which were significant but at different levels (HILS1 at $p < 0.001$, HILS2 at $p < 0.01$, HILS3 at $p < 0.01$, HILS4 at $p < 0.01$ and HILS at $p < 0.05$). The standardized regression weights on SWBS for each item ranged from 0.62 to 0.85 for items of SWLS, 0.82 to 0.96 for items of HILS 0.39 to 0.62 for items of NA, and 0.25 to 0.57 for items of PA. The standardized regression weights on specific subscales for their respective items ranged from 0.37 to 0.47 for SWLS, −0.46 to 0.18 for HILS, 0.45 to 0.67 for PA and −0.54 to −0.71 for NA. In this model, we reversed the scores of the NA items because these items could yield negative factor loadings on SWBS (Fig. 1; see also Fig. S10 for the original model with non-reversed NA items in the Supplemental Material). In general, the results from the bifactor model indicated that: (1) the model fit indices were acceptable and the best ones compared with the other tested models in this study, (2) the loadings of HILS' items on the general latent factor (SWBS) were very high and the highest and at the same time the loadings of these items on the specific latent factor (HILS) were very low and the lowest, thus, indicating that the HILS' items contributed highly to the general factor (SWBS) but no to HILS itself, (3) the loadings of items of SWLS on the general latent factor (SWBS) were also high and still high

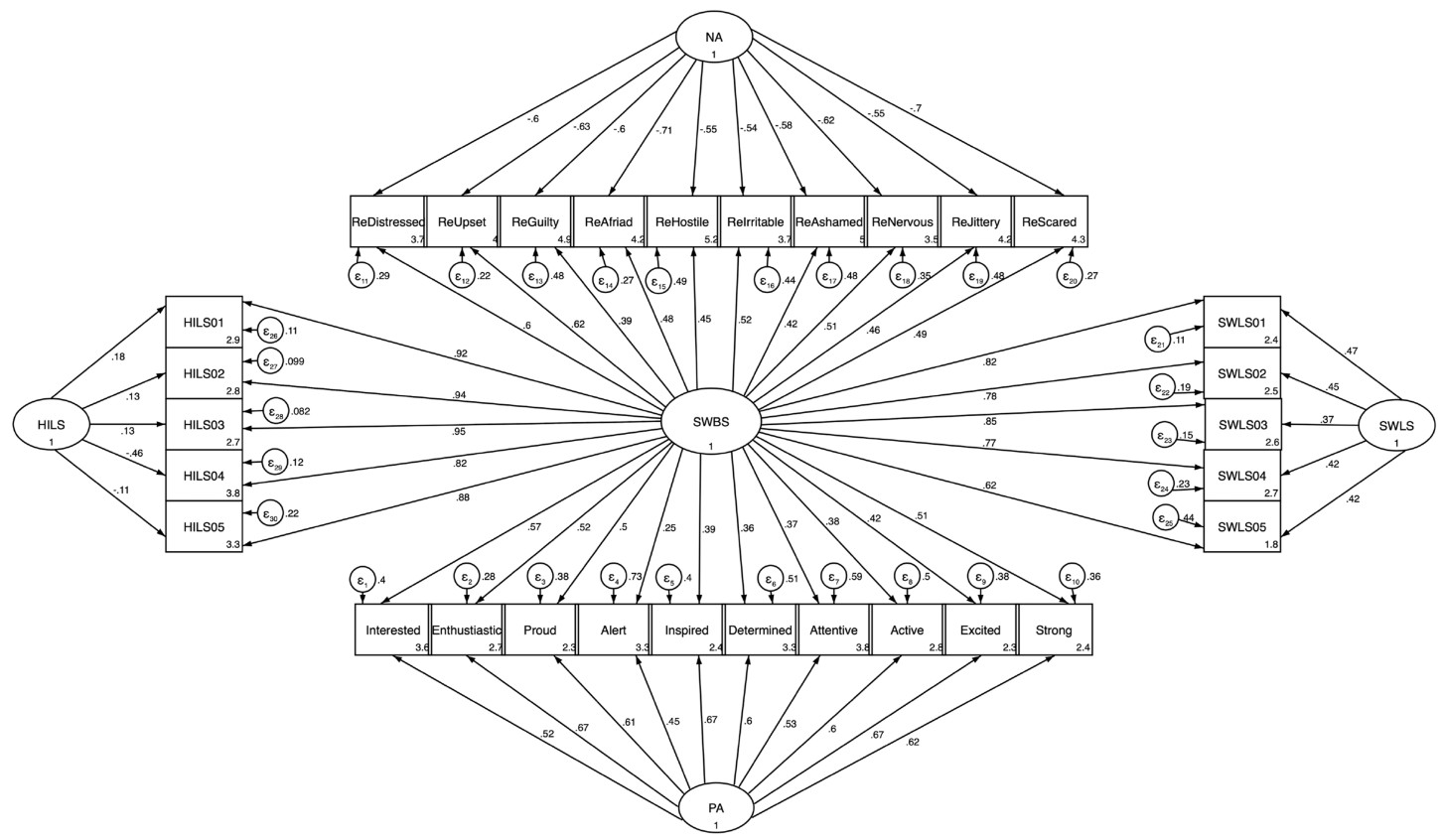

**Figure 1 Structural equation model of bifactor model of SWBS and its specific subscales (SWLS, HILS, NA and PA).** Structural equation model of bifactor model of SWBS and its specific subscales (SWLS, HILS, NA and PA). All paths (from SWBS to each item, and from specific subscales to their items) and their standardized parameter estimates. Chi-square value ($\chi^2$ = 1881.49, df = 401, $p$ < 0.001), Satorra Bentler $\chi^2$ (S–B $\chi^2$ = 1419.90, df = 401, $p$ < 0.001), CFI = 0.91, TLI = 0.90 and RMSEA = 0.07. RMSEA, CFI, and TLI goodness-of-fit statistics are computed using the Satorra–Bentler scaled chi-squared statistic ($N$ = 527).

but moderate for the specific latent factor (SWLS), thus, confirming that the SWLS' items contributed to a higher degree to the general factor (SWBS) compared to the items' contribution to SWLS and (4) the loadings of items of PA and NA on the general latent factor (SWBS) were lower compared with the loadings of these items on their corresponding specific latent factor (PA respectively NA), thus, indicating that the PANAS' items reflected a mixture of general latent structure saturation and specific latent structure saturation, but they still contributed to their respective specific latent factor more than to the general latent factor (SWBS).

In sum, the five HILS items could not clearly and explicitly cover HILS itself as a specific factor (e.g., HILS4 and HILS5 had negative loadings on HILS). Moreover, the multidimensional models seemed to fit better with our data than the unidimensional models. This bifactor structure could capture the nature of the proposed multidimensionality of a biopsychosocial model for subjective well-being that considers all the parts of human health and well-being: physical, psychological, and social. In other words, at least with the instruments used here, we have discerned a biopsychosocial model of subjective well-being that needs to consider all three components as expected,

where the social component is part of the whole but not a part by its own right. Next, we go through all details regarding the bifactor model.

## Factors' eigenvalues for the bifactor model

The results showed that general factor (SWBS) had a large eigenvalue 11.47, that two of the subscales had eigenvalue greater than one (3.58 for PA and 3.73 for NA), and that HILS and SWLS had eigenvalues that were less than one (0.91 for SWLS and 0.29 for HILS). These eigenvalues together with the results presented above indicated that most of the variance in total scores in this given bifactor model could be attributed to the SWBS as a general latent factor and to the PA and NA as two specific latent factors. Moreover, the eigenvalue for SWLS indicated that SWLS might be interpreted as a specific latent factor, while HILS was clearly not a specific latent factor.

## Reliability coefficients for the bifactor model

The results indicted that *Omega total* was 0.79 (Cronbach's alpha was 0.96 for SWBS, ranging from 0.92 to 0.95 for the subscales). Thus, both the general latent factor (SWBS) and the specific latent subscales (SWLS, PA, NA, and HILS) could explain 79% of the total variance of all 30 items in our bifactor model. In other words, only 21% of the variance was estimated due to random error (uniqueness) and therefore not explained by the bifactor model. *Omega hierarchical* was 0.64, thus, 64% of the variance in total scores could be attributed to SWBS as a single general latent factor. Importantly, a value above 0.50 regarding *Omega hierarchical* indicates a broad general latent trait (*Reise, Bonifay & Haviland, 2013*). Moreover, the value of *Omega hierarchical* indicated also that 81% (0.64/0.79 = 0.81) of the common variance in total scores belonged to a single general latent factor (i.e., SWBS). About 15% (0.79−0.64 = 0.15) of the variance in total scores was estimated to be due to all specific latent traits (SWLS, HILS, NA and PA). In other words, for the bifactor model, 19% of reliable common variance (i.e., the reliable variance for both the general factor and subscales) in total scores could be attributed to the specific subscales (i.e., SWLS, HILS, NA and PA).

On the subscale level, *Omega hierarchical subscales* were 0.22 for SWLS, 0.00 for HILS[5], 0.48 for PA, and 0.49 for NA, while *Omega subscales* were 0.94 for SWLS, 0.98 for HILS, 0.72 for PA, and 0.81 for NA. This indicated that, in the bifactor model, when the reliable variance of the single general latent factor (SWBS) was removed from the subscales, the reliability of the specific subscales was largely reduced due to the large effect of the general latent factor (SWBS). This means, for example, that the extremely high value of *Omega subscale* for HILS (0.98) was caused by the extremely high positive loadings between the HILS' items and the general factor (i.e., SWBS; 0.92 for HILS1, 0.94 for HILS2, 0.95 for HILS3, 0.82 for HILS4, and 0.88 for HILS5). Thus, the 0.00 value of *Omega hierarchical subscale* for HILS was caused by the small and negative loadings between the HILS' items onto the HILS subscale (0.18 for HILS1, 0.13 for HILS2, 0.13 for HILS3, −0.46 for HILS4, and 0.18 for HILS5) and probably also influenced by both positive and negative loadings, thus, canceling each other out (*Rodriguez, Reise & Haviland, 2016*). Based on these results, we argue that the high loadings between the HILS' items and HILS

[5] The negative loadings regarding HILS4 "I accept the various conditions of my life" and HILS5 "I fit in well with my surroundings" on their corresponding subscale made us use absolute values of factor loadings. Then we calculate the sum of these loadings to calculate omega indices. The result didn't differ, except with small changes regarding *Omega hierarchical subscale* for HILS (changed from 0.00 to 0.05), *Omega hierarchical* (changed from 0.64. to 0.63) and *Omega general for subscale* (changed from 0.98 to 0.93). See Table S4.

as a single latent factor (See Fig. S2) belong to the latent trait of SWBS rather than the latent trait of HILS. In other words, the high loadings between the HILS' items and the HILS single factor reflect merely the effects of the shared variance caused by the repeated item content, rather than each item's relation with HILS as a single factor. Thus, the HILS' items do not reflect any shared variance or maybe just minimal variance of HILS as a subscale.

The *Omega hierarchical subscales* and *Omega subscale* for SWLS were 0.22 and 0.94 respectively. This indicated that the SWLS items tended to load higher on the general latent factor (SWBS) than on their corresponding subscale (SWLS). In other words, only small reliable variance on the subscale (SWLS) level remained when the general latent factor was controlled for. This subscale reliability is mostly attributable to individual differences on the general latent trait, which had an *Omega general for subscale* 0.72 (0.94−0.22 = 0.72). The values of *Omega hierarchical subscales* for PA (0.48) and for NA (0.49) and *Omega subscale* for PA (0.72) and for NA (0.81) showed that the PA and NA items tended to load higher on their corresponding subscales than on the general latent factor (SWBS). PA and NA's reliability of a subscale scores were still high after removing the reliable variance of the general factor. Additionally, both values of *Omega general for subscale* (0.25 for PA and NA.32) and factor loadings on their corresponding subscale (ranging for PA from 0.25 to 0.57 and for NA from 0.39 to 0.62) showed that items of PA tended to load lower on the general latent factor compared with items of NA. See the Supplemental Material for the details. In this context, there is extensive evidence that PA and NA are best thought as two distinct and dissociable factors and some evidence that they have different heritability, NA showing the strongest genetic influence (for a review see *Cloninger & Garcia, 2015*). Thus, probably explaining the differences found here.

## Explained common variance for the bifactor model

The result indicated that 57% of the common reliable variance (i.e., reliable variance due to single general factor and group factors) was explained primarily by the single general factor (SWBS). In other words, 43% of common reliable variance in all 30 items in our bifactor model was spread across the subfactors (i.e., SWLS, HILS, PA and NA). In general, high value of ECV indicates that the model has a strong general latent dimension rather than latent subdimensions. Thus, our results indicated that both the single general factor and the group factors contributed to the common reliable variances in our bifactor model. The high factor loadings on SWLS and HILS might have caused the high ECV values (0.57 = 57% of common reliable variance) and the high $\Omega H$ (0.64 = 0.64% of variance in total score), whereas the high factor loadings on NA and PA might have caused 43% of the common reliable variance that was spread across the subscales. Thus, indicating the presence of a multidimensional structure that contains both a single general factor and subfactors. Indeed, tests of ECV calculations using various datasets suggests that data with ECV < 0.70 are multidimensional and should therefore be decomposed into multiple scales (ECV > 0.90 indicates unidimensionality; *Quinn, 2014*).

## Item explained common variance for the bifactor model

I-ECV is suggested as a useful index to identify the percentage of expected common variance by the general latent factor at the item level. The result showed that the means of I-ECV were 0.76 for SWLS, 0.93 for HILS, 0.34 for PA, and 0.40 for NA. The values concerning HILS (0.93) and SWLS (0.76) showed clear strong effects sizes for the loadings between the HILS and SWLS items and the general factor, whereas the lower values concerning PA (0.34) and NA (0.40) showed clearly that the majority of the percentage of expected common variance was explained by the loadings between the PA and NA items and their corresponding subfactors rather than the loadings between the PA and NA items and the general factor. I-ECV can also be applied to get a broad sense of the extent to which items in a model can be selected to represent only a unidimensional factor as a good indicator of the general dimension. Items that have values above 0.80 or 0.85 are suggested to largely reflect a general dimension as a broad construct (*Stucky & Edelen, 2014*). Thus, SWLS and HILS had a tendency for SWBS rather than their respective original concepts, whereas PA and NA had a tendency for their respective original concepts rather than SWBS. However, some items contributed equally to both the general factor and their respective subfactors (i.e., I-ECV values: 0.50 for "Distressed", 0.49 for "Upset", and 0.48 for "Irritable"). Moreover, the item "Alert" and the item "Attentive" had the largest error variances (uniqueness) 0.74 and 0.58 respectively. Thus, 74% and 58% of the variance in these items was unique and did not share reliable variance with any other items in our bifactor model. So, these items should be removed or modified (Table 1).

## Traditional average scores and factor scores between models

We calculated the scores for all scales (SWBS, SWLS, HILS, NA and PA) using the traditional average score approach and the different models' factor scores (i.e., unidimensional model, correlated factors model, second order factor model and bifactor model). See Tables 2 and 3 for the details. Additionally, we tested and described the correlations among all the different scores of each scale using Pearson's correlation coefficient. The result showed that all the correlations were very high and significant ($p < 0.01$) regarding the scores computed using the traditional average score, the unidimensional model, the correlated factors model, and the second order factor model. The correlations ranged between 0.99 and 1.00 for SWLS, from 0.99, to 1.00 for HILS, from 0.99 to 1.00 for PA, from −1.00 to 0.99 for NA, and from 0.91 to 0.99 for SWBS. Then we compared the score distributions for all different models in our study. The bifactor model had a clear tendency to fit normal distribution better than all the other models (Tables 2 and 3). See also the Supplemental Material for the details (Figs. S11–S15). In general, these results confirmed that there are no differences within scores for each of the scales between these modes, except for the scores regarding the bifactor model. These results suggest that the loadings of each item within each scale did not vary across the following models: unidimensional model, correlated factors model and second order factor model. However, there were small differences among scores of SWBS regarding the traditional average score approach, the unidimensional model scores, and the second order factor model scores. These small differences might suggest that not all items could measure features of SWBS equally. In other words, the traditional

**Table 1 Standardized loadings factor of bifactor confirmatory factor analysis for the general latent trait (Subjective Well-Being, SWBS) and its specific latent traits (Satisfaction with Life, SWLS, Harmony in Life, HILS, Positive Affect, PA, and Negative Affect, NA).**

| Items | SWBS | SWLS | HILS | PA | NA | h2 | u2 | p2 |
|---|---|---|---|---|---|---|---|---|
| In most ways my life is close to my ideal | 0.82 | 0.47 | | | | 0.89 | 0.11 | 0.75 |
| The conditions of my life are excellent | 0.78 | 0.45 | | | | 0.81 | 0.19 | 0.75 |
| I am satisfied with my life | 0.85 | 0.37 | | | | 0.86 | 0.14 | 0.84 |
| So far I have gotten the important things I want in life | 0.77 | 0.42 | | | | 0.77 | 0.23 | 0.77 |
| If I could live my life over, I would change almost nothing | 0.62 | 0.42 | | | | 0.56 | 0.44 | 0.69 |
| My lifestyle allows me to be in harmony | 0.92 | | 0.18 | | | 0.88 | 0.12 | 0.96 |
| Most aspects of my life are in balance | 0.94 | | 0.13 | | | 0.90 | 0.10 | 0.98 |
| I am in harmony | 0.96 | | 0.13 | | | 0.94 | 0.06 | 0.98 |
| I accept the various conditions of my life | 0.82 | | −0.46 | | | 0.88 | 0.12 | 0.76 |
| I fit in well with my surroundings | 0.88 | | −0.11 | | | 0.79 | 0.21 | 0.98 |
| Interested | 0.57 | | | 0.52 | | 0.60 | 0.40 | 0.55 |
| Enthustiastic | 0.52 | | | 0.67 | | 0.72 | 0.28 | 0.38 |
| Proud | 0.50 | | | 0.61 | | 0.62 | 0.38 | 0.40 |
| Alert | 0.25 | | | 0.45 | | 0.27 | 0.74 | 0.24 |
| Inspired | 0.39 | | | 0.67 | | 0.60 | 0.40 | 0.25 |
| Determined | 0.36 | | | 0.60 | | 0.49 | 0.51 | 0.26 |
| Attentive | 0.37 | | | 0.53 | | 0.42 | 0.58 | 0.33 |
| Active | 0.38 | | | 0.60 | | 0.50 | 0.50 | 0.29 |
| Excited | 0.42 | | | 0.67 | | 0.63 | 0.37 | 0.28 |
| Strong | 0.51 | | | 0.62 | | 0.64 | 0.36 | 0.40 |
| Distressed | 0.60 | | | | −0.60 | 0.72 | 0.28 | 0.50 |
| Upset | 0.62 | | | | −0.63 | 0.78 | 0.22 | 0.49 |
| Guilty | 0.39 | | | | −0.60 | 0.51 | 0.49 | 0.30 |
| Afraid | 0.48 | | | | −0.71 | 0.73 | 0.27 | 0.31 |
| Hostile | 0.45 | | | | −0.55 | 0.51 | 0.50 | 0.40 |
| Irritable | 0.52 | | | | −0.54 | 0.56 | 0.44 | 0.48 |
| Ashamed | 0.42 | | | | −0.58 | 0.51 | 0.49 | 0.34 |
| Nervous | 0.51 | | | | −0.62 | 0.64 | 0.36 | 0.40 |
| Jittery | 0.46 | | | | −0.55 | 0.51 | 0.49 | 0.41 |
| Scared | 0.49 | | | | −0.70 | 0.73 | 0.27 | 0.33 |
| Omega-total ($\Omega Total$) | 0.79 | | | | | | | |
| Omega Hierarchical ($\Omega H$) | 0.64 | | | | | | | |
| Omega subscale ($\Omega S$) | | 0.94 | 0.98 | 0.72 | 0.81 | | | |
| Omega hirerarchial subscale ($\Omega HS$) | | 0.22 | 0.00 | 0.48 | 0.49 | | | |
| Omega general for subscale | | 0.72 | 0.98 | 0.25 | 0.32 | | | |
| ECV | 0.57 | | | | | | | |
| Eigenvalues | 11.47 | 0.91 | 0.29 | 3.58 | 3.73 | | | |

**Note:**
h2, communalities; u2, error variance (uniqueness); p2, item explained common variance (I-ECV). Raw items of the NA are Reversed.

average score approach seems to ignore the possible differences within the items, thus, assuming equal contribution of each item on SWBS. However, the standardized loadings are actually different across the items (e.g., "Alert" had low and the lowest standardized

**Table 2 Descriptive statistics for all the scores using the different methods applied in the study (N = 527).**

| Variables | Minimum | Maximum | Mean | SD | Skewness | Kurtosis |
|---|---|---|---|---|---|---|
| SWLS Traditional average | 1.00 | 7.00 | 4.56 | 1.72 | −0.60 | −0.71 |
| SWLS Unidimensional Model | −3.68 | 2.36 | 0.00 | 1.75 | −0.69 | −0.64 |
| SWLS Correlated Model | −3.88 | 2.41 | 0.00 | 1.75 | −0.71 | −0.57 |
| SWLS Higher Order Factor | −3.88 | 2.39 | 0.00 | 1.75 | −0.71 | −0.56 |
| SWLS Bifactor | −4.11 | 2.51 | 0.00 | 0.79 | −1.12 | 3.43 |
| HILS Traditional average | 1.00 | 7.00 | 5.02 | 1.50 | −0.90 | 0.10 |
| HILS Unidimensional Model | −3.77 | 2.02 | 0.00 | 1.55 | −0.84 | −0.20 |
| HILS Correlated Model | −3.88 | 2.07 | 0.00 | 1.55 | −0.84 | −0.19 |
| HILS Higher Order Factor | −3.86 | 2.08 | 0.00 | 1.55 | −0.83 | −0.20 |
| HILS Bifactor | −3.34 | 3.36 | 0.00 | 0.85 | −0.26 | 1.85 |
| PA Traditional average | 1.00 | 5.00 | 3.31 | 0.90 | −0.19 | −0.47 |
| PA Unidimensional Model | −1.75 | 1.34 | 0.00 | 0.72 | −0.14 | −0.60 |
| PA Correlated Model | −2.55 | 1.93 | 0.00 | 1.02 | −0.16 | −0.60 |
| PA Higher Order Factor | −2.55 | 1.93 | 0.00 | 1.02 | −0.16 | −0.59 |
| PABifactor | −2.95 | 2.62 | 0.00 | 0.94 | −0.07 | −0.22 |
| NA Traditional average | 1.00 | 5.00 | 1.68 | 0.83 | 1.56 | 2.06 |
| NA Unidimensional Model | −0.75 | 3.53 | 0.00 | 0.92 | 1.52 | 1.89 |
| NA Correlated Model | −0.72 | 3.22 | 0.00 | 0.84 | 1.50 | 1.82 |
| NA Higher Order Factor | −3.22 | 0.73 | 0.00 | 0.84 | −1.50 | 1.81 |
| NA Bifactor | −2.86 | 3.64 | 0.00 | 0.95 | 0.96 | 1.60 |
| SWBS Traditional average | −2.07 | 1.11 | 0.00 | 0.66 | −0.74 | 0.05 |
| SWBS Unidimensional Model | −4.59 | 2.43 | 0.00 | 1.62 | −0.80 | −0.12 |
| SWBS Higher Order Factor | −1.59 | 0.88 | 0.00 | 0.62 | −0.80 | −0.23 |
| SWBS Bifactor | −2.65 | 1.34 | 0.00 | 0.99 | −0.87 | −0.02 |

Note:
Subjective Well-Being (SWBS) traditional average = A simple average of standardized scores of 30 items including reversed items of Negative Affect (NA), Satisfaction with Life Scale (SWLS) traditional average (simple average of raw scores of 5 items), Harmony in Life Scale (HILS) traditional average (simple average of raw scores of 5 items), NA traditional average (simple average of raw scores of 10 five items), Positive Affect (PA) traditional average (simple average of raw scores of 10 items). Reversed items of NA were included in SWBS Unidimensional model, Higher order factor and Bifactor.

loading 0.29). The small loadings of items on SWBS caused the factor scores to be undervalued, whereas, the high loadings of items influenced factor scores for SWBS to a higher degree. These small differences could be also caused by the indirect relationship between SWBS and each item in the second order factor model, whereas, the relationship between SWBS and each item were direct in all the other models. In other words, the complicated second order factor model, with indirect relationships between SWBS and the 30 items, was less clear with regard to the item variance explained by SWBS, due to the mediating sub-paths across SWLS, HILS, NA and PA. Moreover, the SWBS score contained more items (30 items) compared with any of the subscales (e.g., SWLS had only 5 items), so the differences concerning the number of loadings might have a more dramatic effect on the SWBS scores than on the scores of the subscales. Finally, the

**Table 3 Correlations of SWBS and its subscales via different models in the study ($N = 527$).**

| Variables | 1 | 2 | 3 | 4 | 5 |
|---|---|---|---|---|---|
| SWLS Traditional average 1 | | | | | |
| SWLS Unidimensional Model 2 | 0.991** | | | | |
| SWLS Correlated Model 3 | 0.987** | 0.997 | | | |
| SWLS Higher Order factor 4 | 0.987** | 0.997** | 1.00** | | |
| SWLS Bifactor 5 | 0.535** | 0.527** | 0.461** | 0.460** | |
| HILS Traditional average 1 | | | | | |
| HILS Unidimensional Model 2 | 0.991** | | | | |
| HILS Correlated Model 3 | 0.991** | 0.998** | | | |
| HILS Higher Order Factor 4 | 0.992** | 0.998** | 1.00** | | |
| HILS Bifactor 5 | −0.012 | 0.107* | 0.095* | 0.094* | |
| PA Traditional average 1 | | | | | |
| PA Unidimensional Model 2 | 0.994** | | | | |
| PA Correlated Model 3 | 0.992** | 0.999** | | | |
| PA Higher Order Factor 4 | 0.992** | 0.999** | 0.99** | | |
| PA Bifactor 5 | 0.830** | 0.826** | 0.796** | 0.796** | |
| NA Traditional average 1 | | | | | |
| NA Unidimensional Model 2 | 0.996** | | | | |
| NA Correlated Model 3 | 0.994** | 0.999** | | | |
| NA Higher Order Factor 4 | −0.994** | −0.999** | −1.000** | | |
| NA Bifactor 5 | 0.790** | 0.789** | 0.762** | −0.763** | |
| SWBS Traditional average 1 | | | | | |
| SWBS Unidimensional Model 2 | 0.953** | | | | |
| SWBS Higher Order Factor 3 | 0.912** | 0.992** | | | |
| SWBS Bifactor 4 | 0.890** | 0.976** | 0.989** | | |

**Notes:**
* $p < 0.05$.
** $p < 0.01$.
Subjective Well-Being (SWBS) traditional average = A simple average of standardized scores of 30 items including reversed items of Negative Affect (NA), Satisfaction with Life Scale (SWLS) traditional average (simple average of raw scores of 5 items), Harmony in Life Scale (HILS) traditional average (simple average of raw scores of 5 items), NA traditional average (simple average of raw scores of 10 five items), Positive Affect (PA) traditional average (simple average of raw scores of 10 items). Reversed items of NA were included in SWBS Unidimensional model, Higher order factor and Bifactor.

results regarding the bifactor model were very different. The correlations within each scale' scores from the other models and the bifactor model scores ranged from 0.46 ($p < 0.01$) to 0.54 ($p < 0.01$) for SWLS, from −0.01 (*ns*) to 0.11 ($p < 0.05$) for HILS, from 0.80 ($p < 0.01$) to 0.83 ($p < 0.01$) for PA, from −0.76 ($p < 0.01$) to 0.79 ($p < 0.01$) for NA, and from 0.89 ($p < 0.01$) to 0.99 ($p < 0.01$) for SWBS.

## DISCUSSION

In the present study, we added judgements of one's social interactions (harmony in life) to judgements of one's emotional reactions (bio) and judgments of one's life satisfaction (psycho), and used CTT to investigate different factorial models of our theorized biopsychosocial general subjective well-being factor and its specific sub-factors. In sum,

(a) the traditional average score approach consider the items as unweighted items, thus, assuming equal contribution from each item in the construct, whereas factorial models (e.g., the bifactor model or any of the other models used here) allow for the loading of each item to contribute as weighted items, (b) in the traditional average score approach and the unidimensional model, the items are influenced only at the single factor level, whereas, in the bifactor model, the items are influenced and determined by both the single general factor and the subfactor level, (c) the correlated factors model consisted of only four interrelated subfactors, whereas the bifactor model consisted of one general factor and four specific subfactors where no orthogonal correlations were assumed among these factors, and (d) the second order factor model could not clearly describe the effect of general latent factor (SWBS) on each item because the first order scales (SWLS, HILS, NA and PA) mediated this effect, whereas the bifactor model had only a direct pathway from SWBS onto each item, so the association between general latent factor and each item could be freely estimated and was not dependent on any mediating sub-paths.

Using the bifactor model, we were able to clearly separate the influence from the broad dimension (SWBS) and each subdimensions (SWLS, HILS, NA and PA) on each item. The bifactor model was successful in covering the general latent dimension, and allowed us to calculate the scores for specific latent dimensions in a more precise and purified form without the weakness of the other models (e.g., indirect correlations, unweight items effect). In addition, the bifactor model showed large differences in score distributions, while the other models showed almost equal score distributions. So, we suggest that the bifactor model could clearly describe and support the notion of a biopsychosocial model.

Indeed, the multidimensional models fit the data better than the unidimensional models. Both general latent factor (SWBS) and specific latent subscales could explain 79% of the total variance in our bifactor model. This is a clear indication of the presence of a multidimensional structure that contains both a general factor and subfactors. The SWLS items tended to load higher on the general latent factor (SWBS) than on their corresponding subscale (SWLS). In other words, only very little reliable variance on the SWLS subscale remained when the general latent factor was removed, and this reliability of subscale was mostly attributable to individual differences on the general latent trait rather than to individual differences on the specific latent trait. The PA and NA items reflected a mixture of general latent structure saturation and specific latent structure saturation, but they contributed more to their respective specific latent factor than to the general latent factor (SWBS). Moreover, in the bifactor model, 74% of the variance for "Alert" and 58% of the variance for "Attentive" were error variances (uniqueness) and did not share reliable variance with any other of items within the model. So, we recommend to remove or replace these items following the recommendations found elsewhere—for instance, these very same items, "Alert" and "Attentive", provided lesser information when Item Response Theory was applied using a different sample (*Nima et al., 2020*). In this context, some theories suggest that besides the dimension of positive and negative, there is a high and low activation dimension (*Russell, 1980*; *Russell & Feldman Barrett, 1999*).

That is, emotions are categories that are vertically organized as a fuzzy hierarchy and horizontally organized as part of a circumplex (*Russell & Feldman Barrett, 1999*). In this circumplex, "Alert" and "Attentive" are categorized on the highest point of the high activation dimension and at the lowest point of the positive dimension. It is therefore plausible to suggest that it is this specific feature (i.e., low positive and high activation types of emotions) in these items what causes the problems highlighted using both CTT here and Item Response Theory in other studies (e.g., *Nima et al., 2020*).

Furthermore, the HILS items contributed to a very high degree to the general factor (SWBS), but they did not contribute to HILS itself. Some items had even negative loadings with the HILS subscale within the bifactor model. For instance, some of the HILS' items contained words that do not directly address balance, adaptation or harmony, which are keywords people use to refer to the sense of harmony (*Kjell et al., 2016*), or referred to themes that could be easily misinterpreted. Indeed, harmony is a construct that indicates how people think about their life in relation to the world around them—a process that involves acceptance and adaptation in order to bring balance to one's life (Garcia, Nima, Granjard & Cloninger, 2020, under editorial evaluation). For example, in HILS4: "I accept the various conditions of my life" the keyword is "accept" and in HILS5 "I fit in well with my surroundings" the keyword is "fit". In HILS5, the word "surrounding" might be experienced as unclear or vague, is the participant supposed to think about only physical near places or even family, work, friends, nature? Also, in this line, regarding HILS4, is it necessary for harmony to only "accept" the conditions of one's life even if they are bad conditions? Or is the process of harmony comprised of both acceptance and then adaptation in order to find balance or homeostasis? We find it plausible to suggest that the lack of the process of adaptation and homeostasis as a part of a harmonious life in HILS4 is what generated the negative loadings for this item. Certainly, we may accept bad situations, bad surroundings, and bad conditions that are outside of our possibility to change or avoid, but if we are flexible, resourceful, kind and self-aware, we would be able to do many things to adapt by improving ourselves, others or moving away from the bad situation (*Cloninger, 2004*). So, we conclude that such items should be removed or modified accordingly and then tested again to see if they support or not our theoretical model.

Moreover, despite the fact that the NA scale tended to contribute more to itself than to the SWBS, some NA items (i.e., "Distressed", "Upset" and "Irritable" with I-ECV values = 0.50, 0.49 and 0.48, respectively) contributed equally to both the general factor and the NA subfactor. That being said, we suggest that the bifactor model was useful and an appropriate methodological alternative that allowed us to calculate purified scores concerning general and specific latent factors compared with the traditional average score approach and any of the other models (for benefits and limitations of bifactor analysis see *Bornovalova et al., in press*). The bifactor model could capture the nature of multidimensionality as suggested by our biopsychosocial approach towards subjective well-being, which considers all parts of human health and well-being: physical, psychological, and social.

## Limitations, strengths, and final remarks

One of the limitations with factorial models is that they are heavily influenced by the sample characteristics, such as, number of participants (*Nima et al., 2020*). However, according to *Thompson (2004)* if the factors are defined by four or more measured variables with structure coefficients <0.60, then the sample size is not important. Moreover, if the factors are defined with 10 or more structure coefficients each around 0.40, then the sample size should be at least 150. Any sample size over 300 is considered adequate (see for example *Comrey & Lee (1992)*, who suggest that 50 cases is very poor, 100 is poor, 200 is fair, 300 is good, 500 is very good, and 1,000 or more is excellent) and 10 observations per variable is suggested as a minimum necessary to avoid computational difficulties. In other words, we suggest that our sample size ($N = 527$) with high loadings in most of the items is adequate and even very good for the analyses conducted. That being said, despite the fact that we used different factorial models in order to determine if both a general factor and specific sub-factors contribute to a biopsychosocial model of subjective well-being, we did not use Measurement Invariance analyses (using, for example, SEM). We suggest that future studies use Measurement Invariance analyses to determine whether structural factor loadings, intercepts, residual variances and model fit indexes are similarly/equivalent across multiple populations/groups (e.g., cultures, ethnicities, countries, age, and gender) and/or over different occasions. Another limitation is that we tested convergent and discriminant validity only between the scales of the model, so we recommend that future studies should test convergent and discriminant validity using the scores of these scales that where generated by the bifactor model and other factors that are important for subjective well-being (e.g., personality, psychological well-being).

Although the fit indexes of the bifactor model were better, compared with all the other models, they were still not excellent. More theoretical and practical evidence is required to clarify the psychometric properties of SWLS, HILS, PA, and NA when they are suggested as part of a whole or a biopsychosocial model of subjective well-being. For example, the social construct proposed here, harmony in life, is operationalized with a relatively new instrument that needs further development (*Kjell & Diener, 2020*). Furthermore, the concept of a social part of subjective well-being needs further discussion. One could argue that the concept of resilience, the ability to cope successfully in adverse circumstances, might be a plausible contender for the title of the social part of subjective well-being. Although we agree with this to some point, we argue that resilience is the combination of different personality traits (cf. *Eley et al., 2013*), rather than a construct equal to life satisfaction or positive and negative affect that are judgements of biological emotional reactions and judgements of one's life in relation to a psychological self-imposed ideal, respectively. In contrast, harmony is a construct that indicates how people think about their life in relation to the self, others, and the world around them—a process that involves acceptance and adaptation in order to bring balance to one's life. Since our results suggest a clear tendency for multidimensionality, rather than unidimensionality, further studies should replicate our findings using bifactor

analysis, but also *Multidimensional Item Response Theory*. Furthermore, we only used the most common instrument to operationalize each subjective well-being component. There are however different well-validated scales that can be used in future studies (for a compilation see, for example, *Lopez & Snyder (2003)*). For instance, although it has been very widely applied, the PANAS may not be such a good measure for operationalizing the affective component of a biopsychosocial model of subjective well-being, since it showed evidence of bad fit in the unidimensional models (see for example *Rice & Shorey-Fennell (2020)*, who suggest the Scale of Positive and Negative Experiences developed by *Diener et al. (2010)*). That being said, we suggest that the bad fit of the PANAS only reflects the multidimensional nature of affect, as measured by the PANAS. This suggestion was partially confirmed by the bifactor model in which PA and NA showed both clear general and specific tendencies.

One important caveat here is that most of the times, subjective well-being is measured through self-reports or having people giving their recollection of emotions, life satisfaction and even harmony in life. As such, all components are then cognitive in nature and should therefore be seen as parts of a cognitive whole that is influenced by internal states and traits and external situations (*Schwartz & Strack, 1999*). Nevertheless, despite the fact that the components are psychological representations, they are also distinctive within a biopsychosocial perspective. The affective component, for example, is a psychological characteristic representing the emotional experience of a person; an experience that depends on their nervous system, which is biological in nature. Likewise, harmony in life is also a cognitive phenomenon that is an evaluation of one's inner state and social interactions between the self, others and the world around. We suggest that the lack of a biopsychosocial perspective probably explains the confusion in the literature with regard to harmony being seen as a complement to the cognitive part of subjective well-being and not as an own part of the whole subjective well-being concept. Something that was clearly showed in our analyses. In other words, the overarching biopsychosocial model of subjective well-being proposed here is purely cognitive in nature. Since subjective well-being is cognitive in nature, at least as measured in most of the literature, we propose that the biological part of this cognitive whole is affect, the psychological part is life satisfaction, and the social part is harmony in life. By measuring all three parts, we get a holistic view of what makes people flourish and resilient. After all, besides optimal conditions for human potential (i.e., flourishing), we need also to understand what makes people adapt and remain healthy, happy, and fulfilled in the face of current world challenges (*Cloninger, 2013a*, *2013b*).

### Funding

The authors received no funding for this work.

### Competing Interests

The authors declare that they have no competing interests. Danilo Garcia is the Head of Research of the Blekinge Center of Competence, which is the Region Blekinge's research
and development unit. Ali Al Nima is the main statistician at the center. The Center works on innovations in public health and practice through interdisciplinary scientific research, community projects, and the dissemination of knowledge in order to increase the quality of life of the habitants of the county of Blekinge, Sweden. Kevin M. Cloninger is the CEO of Anthropedia Foundation. The Anthropedia Foundation is an educational non-profit organization that teaches individuals, professionals, and nonprofits ways to cultivate mental health and well-being in order to decrease rates of lifestyle- and stress-related illness.

## Author Contributions

- Ali Al Nima conceived and designed the experiments, performed the experiments, analyzed the data, prepared figures and/or tables, authored or reviewed drafts of the paper, and approved the final draft.
- Kevin M. Cloninger performed the experiments, authored or reviewed drafts of the paper, and approved the final draft.
- Franco Lucchese performed the experiments, authored or reviewed drafts of the paper, and approved the final draft.
- Sverker Sikström performed the experiments, authored or reviewed drafts of the paper, and approved the final draft.
- Danilo Garcia conceived and designed the experiments, performed the experiments, prepared figures and/or tables, authored or reviewed drafts of the paper, and approved the final draft.

## Human Ethics

The following information was supplied relating to ethical approvals (i.e., approving body and any reference numbers):

This work was exempt from ethical review under Swedish law.

## Data Availability

Raw data are available as a Supplemental File.

## Supplemental Information

Supplemental information for this article can be found online at http://dx.doi.org/10.7717/peerj.9193#supplemental-information.

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
