# Peer review of "Validation of a general subjective well-being factor using Classical Test Theory"

_PeerJ, doi:10.7717/peerj.9193_

## Round 0.1 · original submission · Major Revisions

With the Reviewers comments in hand, I recommend a major revision of the manuscript.

In fact, both reviewers are overall positive about your work.

However, they both raise some methodological and analytical issues, and they both provide some constructive suggestion on how to improve your analyses, correct some terminological issues and avoid redundancy in reporting the results. They both recommend also some alternative and/or additional analyses that may improve your manuscript (e.g., using parallel analysis in you EFA, adding measurement invariance analyses).

Furthermore, Reviewer #2 raises some theoretical issues that need to be addressed as well.

Overall, I concur with the Reviewers that your manuscript is well conducted and provides an important contribution to the field of subjective well-being measurement. However, I agree with them that there is a substantial room of improvement.

Best,
MTL

·

Basic reporting

Some phrasings seem not quite right (e.g. l.406: “Because the large different between” should be “Because of the large differences…”). Native speaker corrections could help resolve these small issues.

The article is structured in the typical way and easy to follow. Raw data is supplied and well-labeled. Tables and Figures are good; only Figure 1 in the main manuscript appears to be low-resolution for some reason, which makes it harder to read. Most of the parameter values can not be made out clearly.

Experimental design

Ethical approval was not given by any ethics committee, but should not be relevant in a survey study.

Validity of the findings

no comment

Additional comments

Thank you for the opportunity to review this exciting manuscript.
It is obvious from the submitted documents that the authors took great care in providing a thorough and in-depth investigation of the issue at hand.

• Throughout the manuscript, (non-greek) statistical indicators are sometimes italicized, sometimes they aren’t. This should be done consistently.
• l.119 – Here the authors cite “Li, 2008ab”, but only one citation by Li (2008) is included in the manuscript. Thus, no subsetting with letters is necessary.
• l.216f – Shouldn’t this be “neither agree nor disagree”?
• l.243 – The authors write that they conducted comparisons with the saturated model. However, this is probably not correct, from my understanding. With some fit indices the proposed model is compared to the null (or independence) model but this is the exact opposite of the saturated model. Also, the robust χ² value is utilized not only in these comparative fit indices but in all calculations that follow.
• l.251 – “This” instead of “These”
• l.266 – Add “robust” before “maximum likelihood”, also Satorra & Bentler (2001) should be cited if their procedure is used.
• l.281f – It is unclear to me why the authors report both α and ω, especially after describing the flaws of α for an entire page, but this is not wrong per se.
• l.307f – ω is not printed correctly
• l.308f – The authors describe ω as an estimate of item variance “that belongs to a reliable variance for the general latent factor and also for specific subscale(s)”. In my understanding of the term, “reliable variance” always belongs to a construct being measured (as opposed to the error variance). Thus, the “reliable” seems redundant here when one writes that the variance belongs to a factor.
• l.325 – I am not convinced that scores should be calculated for all estimated models, since some of them evinced relatively bad fit.
• l.349f – The authors should use a modern criterion for the EFA (such as Horn’s Parallel Analysis) instead of the Kaiser-Guttman criterion. Also, testing all of the scales individually “only” provides additional support for the already existing scales (which is always a good thing, but doesn’t really help their present aim).
• l.353 – PCA is not EFA in strict terms (see e.g., https://stats.stackexchange.com/questions/123063/is-there-any-good-reason-to-use-pca-instead-of-efa-also-can-pca-be-a-substitut)
• l.364f – Similarly, they test CFAs for all individual scales. This is again not necessary, IMO, and it takes up a lot of space.
• l.387f – Although it has been very widely applied, the PANAS may not be such a good measure (as the authors themselves note in the Introduction). The PANAS models evince bad fit (Figures S3 and S4), and the full models probably suffered as a result. This should be discussed. Alternative conceptualization of PA and NA could be explored.
• l.398f – RMSEA is known to be inflated with low df models: Kenny, Kaniskan, and McCoach (2014)
• l.403f – In the second run of EFA, again Parallel Analysis would be preferred.
• l.417f – It is not clear to me why the authors present models with and without inverted indicators, as model fit is not affected by this and parameter estimates change in perfectly predictable ways (e.g., a factor loading of .70 becomes -.70, etc.). It would save a lot of space to do just one, preferably the models with inverted items. This applies to Figures S5 and S6, S8 and S9, and the bifactor models.
• l.612f – I don’t fully understand the purpose of extracting all the factor scores, calculating their descriptive statistics and comparing them in correlational analyses. Naturally, all models except the bifactor one will be highly correlated with the traditional scale score. Simlarly, all the descriptive statistics in Table 2 don’t really add to the understanding of the latent constructs.
• l.667f and Conclusion – The authors argue that their findings clearly support the notion of a general well-being factor in the sense of the biopsychosocial model. With regard to some aspects of their results, this may be correct. However, the bifactor (and the second order model) still did not reach acceptable model fit (see e.g., Schmermelleh-Engel, Moosbrugger, & Müller, 2003).
They correctly identify the PANAS as the source for this, but please elaborate further, why this is and how could this be remedied? IMO, this paper is a first step but it is limited because of its reliance on the PANAS, which doesn’t seem to be a fully appropriate measure.
• l.699f makes it sound like SWLS and HILS are as much to blame as PANAS. To me it is pretty obvious where the faults in the model lie (see unifactorial analyses of PA and NA).

Reviewer 2 ·

Basic reporting

Some minor grammatical errors.
Previous literature shoudl be revised and new studies introduced (see review)

Experimental design

New analysis could imporve the paper (see review)
Participant section needs to be improved

Validity of the findings

DIscussion section could be improved by comparing results with the previous studies suggested.

Additional comments

The work entitled “Validation of a general subjective well-being factor using classical test theory" is of great interest. The research is very stimulating; it contains new scientific knowledge and provides comprehensive information for further development of this productive line of research. However, I have some comments to make that should be addressed before I recommend this manuscript for publication
Authors should check for some minor grammatical errors. For instance, this sentence “understood as high levels subjective well-being” should be revised.
There are some paragraphs in the introduction that I do not know if I first, understand and second, agree with them.
“We argue, however, that harmony in life is distinct to life satisfaction and propose instead a biopsychosocial perspective on subjective well-being: affect (bio), life satisfaction (psychological), and harmony in life (social)… Naturally, the evaluation of positive and negative affect is the biological part of subjective well-being, since emotions are derived from our temperament, a part of personality with a strong genetic factor and relatively stable over the life span”

According to this, positive and negative affect are indicators of biological characteristics. Is the PANAS, then, a measure of biological aspects instead of psychological characteristics?
Authors explain the latent structure of the positive and negative affect and introduce different previous studies regarding this. From my point of view, most of the studies they mention are somehow outdated. There is only one study about the structure of positive and negative affect after 2010 (Thompson, 2017). I suggest authors considering recent relevant studies (Galinha, Pereira et al., 2013; Merz, Malcarne et al., 2013; Ortuño-Sierra, Santaren et al., 2015; Ortuño-Sierra, Bañuelos, et al., 2019; Sanmartín, Vicent et al., 2018).
WIth regards to the method, the participants section should include more relevant information. For instance, authors talk about a final sample, how was the initial sample then? Also, what was the age distribution, or socioeconomic level, or study level of the participants?
I would also suggest, reviewing some recent studies for the factor structure of the SWLS, as well as it relationship with the positive and negative affect (Glaesmer, Grande, Braehler, & Roth, 2011; Jovanovic, 2016; Moksnes, Løhre, Byrne, & Haugan, 2014; Ortuño-Sierra, Aritio, Chocarro, Navaridas, Fonseca-Pedrero, 2017).
The authors mentioned that they computed EFA and CFA. In order to do so, a cross validation sample procedure is recommended. Was this performed in the study?. Otherwise, I would recommend doing so, although authors should considering the small sample size. If authors decide not to apply this, it should be mentioned in the limitation section.
Also, the measurence invariance (MI) of the resulting model(/models should be considering as an added analysis that could provide further value to the manuscript.
Finally, authors should consider revising the writing when talking about the study of internal consistency or convergent validity in some parts of the paper. First of all,
validity is not a property of the test but inferences of the scores, and also, there are sources or validity evidences, as it is reflected in the APA standards. Attending to this approach it would be more appropriate to talk about evidences of internal structure or evidences of relation with other variables or external variables. In addition, the reliability is not a characteristic of the test. It is more correct to talk about reliability of the scores or estimation of the reliability of the scores (Prieto & Delgado, 2010).

---

## Round 0.2 · accepted · Accept

I am pleased to inform you that both Reviewers are satisfied with your revision, and therefore your manuscript is accepted for publication on PeerJ.

·

Basic reporting

no comment

Experimental design

no comment

Validity of the findings

no comment

Additional comments

The authors did a great job on the revision and were very meticulous in the changes they implemented.
Personally, I might have made some different decisions here or there, but overall the manuscript is now well-written and up to a high methodological standard. I have no reservations with recommending it for publication.

Best regards,
Bjarne Schmalbach

Reviewer 2 ·

Basic reporting

Authors have addressed all my suggestions. I have no further comments.

Experimental design

The experimental design is now appropriate

Validity of the findings

Conclusions are now well founded and supported by the results

Additional comments

I have no further comments